# Monitoring of Changes in Masticatory Muscle Stiffness after Gum Chewing Using Shear Wave Elastography

**DOI:** 10.3390/jcm10112480

**Published:** 2021-06-03

**Authors:** Cyprian Olchowy, Kinga Grzech-Leśniak, Jakub Hadzik, Anna Olchowy, Mateusz Łasecki

**Affiliations:** 1Department of Oral Surgery, Wroclaw Medical University, 50-425 Wroclaw, Poland; kgl@periocare.pl (K.G.-L.); jakub.hadzik@umed.wroc.pl (J.H.); 2Department of Experimental Dentistry, Wroclaw Medical University, 50-425 Wroclaw, Poland; ania.olchowy@gmail.com; 3Department of Radiology, Wroclaw Medical University, 50-556 Wroclaw, Poland; m.c.lasecki@wp.pl

**Keywords:** shear wave elastography, exercise, stiffness, masticatory muscles

## Abstract

This study aimed to investigate if intensive exercise affects the stiffness of the masticatory muscles measured with shear-wave elastography. The study included a cohort of healthy adults (*n* = 40) aged 40 ± 11 years. In each individual, the stiffness of both the masseter and temporalis muscle was examined three times: at baseline, after 10 min of intensive exercise (chewing gum), and after 10 min of relaxation. Stiffness values (median (IQR)) of both the masseter and temporalis muscle were the lowest at the baseline (11.35 (9.7–12.65) and 10.1 (9.1–10.95)), increased significantly after the exercise (12.5 (11.1–13.25) and 10.3 (10.2–10.52)) and then dropped significantly after 10 min of relaxing (11.75 (9.95–12.6) and 10.2 (9.65–11.9)). The stiffness of the temporalis muscle was significantly lower than that of the masseter muscle. The values of the stiffness of the masseters correlated significantly with the values of the stiffness of the temporalis muscles. Shear wave elastography proved to be a sensitive method for showing changes in the stiffness of the muscles involved in the mastication occurring as a response to the effort, which increased the muscle stiffness. Further research is needed to broaden knowledge on the impact of eating habits and the occurrence of parafunctions on the development of temporomandibular disorders and the condition of masticatory muscles.

## 1. Introduction

The stomatognathic system consists of several components such as skeletal elements (mandible and maxilla), vascular and nervous structures, salivary glands, masticatory muscles, and temporomandibular joints. There are four muscles of mastication, including the temporalis muscle, medial pterygoid muscle, lateral pterygoid muscle, and masseter muscle [1]. All of them are attached to the rami of the mandible of one end and other skeletal structures. The masseter muscle and temporalis muscle are both localized superficially. Easy access makes them easy for examination and visible, so patients link them to masticatory function. The masseter muscle, one of the strongest muscles of mastication, has a rectangular shape and relatively thick belly. It arises from the zygomatic bone and is inserted into the mandible. The temporalis muscle is broad and fan shaped. It arises from the temporal fossa on the skull, but its fibers form a tendon that inserts on the coronoid process of the mandible [1]. The stomatognathic system plays a vital role as it constitutes a functional unit responsible for speech, mastication, and swallowing [2,3].

Malfunctioning of the stomatognathic system requires an interdisciplinary approach. Functional, structural, and anatomical connection of elements of the stomatognathic system with other body structures allows for transferring pathological changes within this system to the surrounding structures. Patients who complain about masticatory muscle pain and hypertrophy, temporomandibular joint pain, sounds in temporomandibular joints (clicking and popping), and abnormal jaw movements are often diagnosed with temporomandibular disorder (TMD) [4,5]. The incidence of TMD varies depending on the type of population and affects 10% to 90% of people [6,7]. The symptoms of TMD are often associated with complaints regarding surrounding structures such as headaches, neck and shoulder girdle pain, and otolaryngological pathologies, including tinnitus [8,9]. A review of the literature carried out by Cuccia and Caradonna suggests that altered tension within the stomatognathic system can have consequences for the whole body [10]. The authors concluded that inaccurate proprioceptive information coming from the stomatognathic system may affect head control and lead to impaired neural control of posture.

Several factors contribute to the functioning of the stomatognathic system and overall oral health maintenance. Parafunctional habits, including daily gum chewing, were found to be significantly associated with the occurrence of signs and symptoms of TMD [11]. Such habits increase the workload of the masticatory muscles, which results in muscle pain and hypertrophy. In this study, we attempted to evaluate how the increased workload posed on the masticatory muscles changes their condition in the experimental settings. This study aimed to investigate how chewing gum intensively affects the stiffness of the masseter and temporalis muscle measured with shear wave elastography. We assumed that those muscles would not be equally affected because the masseter muscle is most often involved in TMD [12,13]. Furthermore, TMD symptoms, such as muscle pain and hypertrophy, concern the masseter muscle to a greater extent than the temporalis muscle [14].

## 2. Experimental Section

The study included a cohort of healthy adults (*n* = 40). All were recruited from January to February 2021. For the study, 40 healthy adult volunteers were enrolled. Only healthy participants were enrolled, that is, those without signs and symptoms suggestive of TMD according to the DC/TMD [15] criteria. The remaining exclusion criteria were as follows: the presence of neuromuscular disorders, malignancy, pain within the masseter muscles, and TMD; a history of TMD; being on muscle relaxants and/or other drugs that can alter the functioning of muscles; any parafunctional oral habit; and pregnancy and breastfeeding.

Participants were examined by a radiologist with seven years of experience in shear wave elastography. Each of the studied individuals was examined three times. Examinations were carried out in the same conditions, with the same chewing gum and settings to diminish the impact of these factors on the outcomes. Measurements were taken in the morning before the first meal. First, the baseline stiffness was measured. Second, each participant was asked to chew a piece of gum with high intensity (approximately 1.5 Hz) for 10 min. Immediately after that, the first stiffness measurement was performed. Third, each participant was asked to relax for 10 min, and immediately after that, the second stiffness measurement was performed. The study design is depicted in Figure 1.

The masseter and temporalis muscle stiffness was measured with the Aixplorer Ultimate device (SuperSonic Imagine, Aix-en-Provence, France) with a high-frequency linear probe SL 18–5 (5–18 MHz). During the shear wave elastography examination, the patients were lying in a supine position. They were asked to remain relaxed and comfortable and to refrain from swallowing. Before the examination, the probe was covered with an ultrasound gel to reduce the air between the probe and the skin, which enabled good visualization. The patient’s tissues were not compressed.

The probe was placed parallel to the longitudinal axis of the masseter muscle in the widest part (the midpoint level) of the masseter muscle belly. Regarding the temporalis muscle, the probe was placed in the temporal area longitudinally to the fibers of the temporalis muscle just above the zygomatic arch. Based on our previous experience, such locations on both muscles can be easily identified and provide repeatable results. However, to ensure the accuracy of repeated measurements, we used a linear skin mark pointing to an exact probe location and orientation. A circular, 4 mm region of interest (ROI) was positioned in the center of the muscle tissue. The ROI of 4 mm was chosen to reflect the size of the masseter and temporalis muscle and avoid the deep and superficial fascia of the muscles. It was located in an area of relatively uniform elasticity as guided by shear wave elastography image and a standard deviation of less than 30% of the mean elasticity value.

During each examination, three measurements were taken, averaged, and recorded. Measurements were validated using an elasticity QA Phantom model 049A (Computerized Imaging Reference Systems, Inc., Norfolk, VA, USA).

The study was conducted according to the Declaration of Helsinki. The study was approved by the Bioethical Committee at the Wroclaw Medical University (KB—633/2020). Participation in the study was voluntary. Each of the participants gave his/her informed consent for participation in the study before its start.

Data were statistically analyzed using MedCalc v. 19.5.3 (MedCalc Software Ltd., Ostend, Belgium). The means and standard deviation were calculated. The Shapiro–Wilk test was used to analyze the distribution of the data. The hypothesis of normal distribution was rejected for all viable except for the second measurement of the masseter muscle. For comparison of the masseter muscle stiffness and the temporalis muscle stiffness, the Wilcoxon test for paired values was employed. For comparisons of stiffness values of the one muscle, the Friedman test with pair-wise comparisons was used. To examine the correlation, Spearman’s rank correlation coefficient values were calculated. In all performed tests, a probability value lower than 0.05 was considered statistically significant.

## 3. Results

All participants who started the study remained in the study and had all three stiffness measurements. The mean age of the study group was 40 ± 11 years (range from 19 to 61 years). The group consisted of 21 men and 19 women. Values of stiffness are presented in Table 1. Comparison of stiffness values between the masseter muscle and temporalis muscle showed consistently lower values for the temporalis muscle (Table 2). For the masseter muscle, the elasticity increased by 1.05 ± 0.53 KPa at the first measurement from the baseline and then dropped by 0.89 ± 0.58 KPa at the second measurement. The elasticity of the temporalis muscle increased by 0.64 ± 0.77 KPa after chewing and dropped by 0.19 ± 0.42 KPa after relaxation.

Next, measurements of the masseter muscle (both right and left) and the temporalis muscle (both right and left) were compared between subsequent measurements. The Friedman test with pair-wise comparisons (Table 3) indicated the presence of significant differences between stiffness measured with shear wave elastography with *p* < 0.0001 for both the masseter and temporalis muscle.

The Spearman’s rank correlation coefficients showed that relationships between masseter muscle stiffness and temporalis muscle stiffness were statistically significant. Correlation coefficients for subsequent measurements were as follows: rho = 0.33 (95% CI 0.121 to 0.514) with *p* = 0.0027 for the baseline, rho = 0.365 (95% CI 0.158 to 0.541) with *p* = 0.0009 for the first measurement, and rho = 0.262 (95% CI 0.0449 to 0.455) with *p* = 0.0189.

## 4. Discussion

Stiffness values of both the masseter and temporalis muscle were the lowest at the baseline, increased significantly after the exercise and then dropped significantly after 10 min of relaxation. The stiffness of the temporalis muscle was consistently lower than that of the masseter muscle (*p* < 0.05 for each comparison). The values of the stiffness of the masseters correlated significantly with the values of the stiffness of the temporalis muscles. The lowest values of stiffness were observed at the baseline, then peaked significantly after exercise, and next dropped again, but remained significantly higher when compared to the baseline values.

Muscle stiffness can increase in several conditions. A common physiological cause if muscle stiffness is exercise. In case of masticatory muscles, it can be due to chewing hard foods or parafunctional habits. Komino and Shiga [16] examined mandibular movements during mastication of chewing of foods with different hardness (gummy jellies containing 6%, 8%, and 10% of gelatin). They found that the masseter muscle activity was smallest for the 6% gelatin jellies and increased as the gelatin content (hardness of jellies) increased, which suggests that hard foods require greater exerted strength. Additionally, the amount of movement become greater for harder foods.

Furthermore, the texture of the food is also important for a chewing pattern. The chewing gum evaluated in this study and gelatin jellies may have a different impact on muscles from other types of food. Cho and Lee [17] assessed the effect of three different foods (soft, sticky, and hard) on the activity of the masseter and temporalis muscle using surface electromyography. Participants of this study were allocated into three groups: TMD, malalignment of the temporomandibular joint, and healthy controls. They found that the pattern of both examined muscles’ activity differed significantly depending on the type of food. The authors concluded that their observation might help to diagnose TMD. The survey distributed through social media in Saudi Arabia suggested that such habits are common [18]. Participants reported daily gum chewing (86%), lip/object biting (59%), clenching (45%), nail-biting (36%), and grinding (32%). In the study conducted on healthcare students [19], TMD coexisted with oral parafunctions in 48.6% of examined students. Moreover, the occurrence is significantly correlated with a TMD diagnosis. Those studies also give some guidance for future research on oral long-term parafunctional habits and their impact on the condition of masticatory muscles.

Stress and anxiety can increase muscle tone as well. Owczarek et al. reported a correlation between intensity of perceived stress and anxiety and the tone of both masseter muscles [20]. Stressful lifestyle is considered to increase masticatory muscle tone, which is in line with findings showing that TMD patients experience greater stress in comparison to the otherwise healthy population [21]. Conditions such as allergic reactions and inflammation may be responsible for temporary increase in stiffness. Muscle tone normalizes after removing a causative factor. Persistent increase in muscle tone can be observed as a side effect of a cholesterol-lowering therapy with statins. Pathologic conditions such as polymyalgia rheumatica, a history of stroke, and cerebral palsy are characterized by increased muscle stiffness; however, more areas are affected, while in TMD, this pathology is limited to masticatory muscles.

An interesting concept of the stomatognathic adaptive motor syndrome as a proper classification for TMD was presented by Douglas et al. [22] They hypothesized that inadequate tooth contact and mandibular position force the mandible to make minimal movements to obtain a better intercuspal position and induce adaptive changes in stomatognathic structures. Signs and symptoms of this condition include those of TMD. Regarding the masticatory muscles, patients can experience pain, hypertonia, fatigue, and weakness, of which hypertonia can be measured with shear wave elastography and shown as increased stiffness. Moreover, such patients complain about difficulties chewing hard food due to pain, fatigue, and weakness of masticatory muscle. A soft diet, as suggested by Douglas et al. [22], may reduce a temporomandibular joint tissues response and, thus, help minimize symptoms. This concept must be further investigated; however, given the wide occurrence of TMD, learning about the impact of eating habits or parafunctions (e.g., habitual gum chewing) on the increased risk of TMD could help in diagnosing and treating TMD. Based on studies on the influence of various factors on mastication, the formulation of recommendations for daily functioning could support standard treatment of TMD.

On the grounds of the stomatognathic adaptive motor syndrome concept [22], including weakness of masticatory muscles as a part of this syndrome, strengthening of masticatory muscles would bring benefits to TMD patients. Indeed, the study by Gavish et al. [23] reported greater pain relief and reduction in the disability score improved significantly in the group undergoing a controlled chewing exercise. The authors concluded that the findings are unequivocal because pain sensation did not change during the chewing test with harder material such as the wax, while the chewing exercise was performed with relatively soft chewing gum. Additionally, Kim et al. [24] reported the benefits from chewing gum in adults aged ≥65 years, such as an increase in occlusal force, greater salivation, and better swallowing function.

Reports from the literature show that shear wave elastography has the potential to be used in the evaluation of the condition of masseter muscle stiffness [25]. Our study showed that chewing gum intensively increases the stiffness of both muscles of the masticatory system significantly. This state was maintained by the end of the relaxing phase, although the masseter muscle had higher values at each time point. Previous studies showed that factors other than effort can change muscle stiffness as well. In other reports, a decreased stiffness as measured by shear wave elastography was reported after applying the massage. Olchowy et al. [26] examined 20 healthy adult volunteers before and after a 30 min massage of the masseter muscle. They observed a drop in stiffness values from 11.46 ± 1.55 KPa to 8.97 ± 0.96 KPa (*p* < 0.0001). In another study, Ariji et al. [27] examined the stiffness of the masseter muscle with sonographic elastography. They enrolled 37 patients with TMD and myofascial pain. The median elasticity index ratios decreased significantly from the baseline to the 3rd session in the responder groups, while non-responders observed a drop in stiffness that was not significant.

This study is not without limitations, such as the relatively small number of subjects involved. However, the study group was relatively homogeneous and consisted of healthy participants. The study evaluated a short-term response to effort followed by a short-term follow-up. Long-term studies would be of much greater value as parafunctional habits can be even lifelong. Furthermore, we did not take into account the subject’s favorite chewing side and chewing pattern, which could alter the stiffness of the muscles on that side. This study examined the effect of one-time intensive mastication of chewing gum by participants who denied having oral parafunctional habits. In this context, further research should focus on people with parafunctional habits with or without TMD and the development of an effective study protocol (e.g., including the evaluation of symmetry) for capturing patterns of changes caused by parafunctional habits.

## 5. Conclusions

Shear wave elastography proved to be a sensitive method for showing changes in the stiffness of the muscles involved in the mastication occurring as a response to the effort. The increase in stiffness was observed in both the masseter and temporalis muscle, and the values of stiffness were correlated significantly. Furthermore, shear wave elastography is objective and non-invasive, which creates a potential of using it in the monitoring of changes in the muscles of mastication in clinical practice. We recommend further research on masticatory muscle stiffness using shear wave elastography.

## Figures and Tables

**Figure 1 jcm-10-02480-f001:**
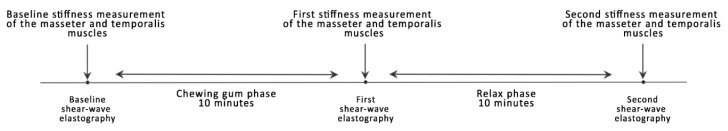
The study design.

**Table 1 jcm-10-02480-t001:** Values of stiffness of the masseter and temporalis muscles.

	Baseline Measurement, Mean (SD), KPa	First Measurement, Mean (SD), KPa	Second Measurement, Mean (SD), KPa
Left masseter muscle	10.99 (2.04)	12.31 (1.38)	11.29 (2.01)
Right masseter muscle	11.01 (2.21)	12.32 (1.65)	11.30 (1.73)
Left temporalis muscle	10.23 (1.23)	10.92 (1.68)	10.65 (1.68)
Right temporalis muscle	10.14 (1.32)	10.73 (1.70)	10.62 (1.71)

Abbreviation: SD, standard deviation.

**Table 2 jcm-10-02480-t002:** Comparison of stiffness measurements as measured with shear wave elastography between the masseter muscle and temporalis muscle.

	Baseline Measurement, Median (IQR), KPa	First Measurement, Median (IQR), KPa	Second Measurement, Median (IQR), KPa
Masseter muscle	11.35 (9.7–12.65)	12.5 (11.1–13.25)	11.75 (9.95–12.6)
Temporalis muscle	10.1 (9.1–10.95)	10.3 (10.2–10.52)	10.2 (9.65–11.9)
Difference in medians	1.25	2.2	1.55
*p*-value	0.0001	0.0001	0.0033

Abbreviation: IQR, interquartile range.

**Table 3 jcm-10-02480-t003:** Comparison of the shear wave elastography results using the Friedman test with pair-wise comparisons.

Median (IQR)	Baseline		1st Measurement		2nd Measurement	
**Masseter Muscle (*p* < 0.0001)**						
Baseline 11.35 (9.7–12.65)			*p* < 0.05		*p* < 0.05	

1st measurement 12.5 (11.1–13.25)					*p* < 0.05	

2nd measurement 11.75 (9.95–12.6)						

**Temporalis Muscle (*p* < 0.0001)**						
Baseline 10.1 (9.1–10.95)			*p* < 0.05		*p* < 0.05	

1st measurement 10.3 (9.95–12.2)					*p* < 0.05	

2nd measurement 10.2 (9.65–11.9)						


## Data Availability

Data is contained within the article.

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
