# Peer review of "Monitoring of Changes in Masticatory Muscle Stiffness after Gum Chewing Using Shear Wave Elastography"

_jcm, 2021, doi:10.3390/jcm10112480_

Round 1
Reviewer 1 Report
Overall comments
The authors present an experimental study that assessed by means of shear wave US the stiffness of masticatory muscles before and after chewing gum.
The paper gives interesting information about the clinical application of this type of measurement in the field of TMD. It might be a valuable reference for other studies.
The paper is clearly structured and easy to follow. However, I think that some content issues should be addressed in order to improve its quality. Namely, I think that the description of the US method principles must be improved, as well as the discussion should be better structured and be more adherent to the presented results and their relevance to the field and their limitations.
Introduction
Lines-69-71 Citations should be added to corroborate these statements
Material and methods
General remark: the order of the experimental session is confused. The recruitment criteria at first, and only then explain the experimental procedure. Consider restructuring.
Lines 75-77 “Examinations were carried out in the same conditions, with the same chewing gum and settings to diminish the impact of other factors on the outcomes.» Maybe the factors that could influence the outcomes should be named
Lines 78-79 There is no clear explanation of the chewing protocol. Any indication on the chewing side? Was kept free? the subject's favorite chewing side was considered? This can be also considered a confounding factor for the results. It would also be interesting to relate the chewing pattern to the chewing side and the changes in the stiffness of the muscles on that side.
Lines 102-103 how was the position of the region of interest defined and how repeatability ensured? Since the measurement is repeated before and after the chewing, the repositioning of the probe and the selection of the ROI are crucial.
Furthermore, the whole analysis of the elasticity is missing. It would be helpful to include how the elasticity is calculated by the instruments and at least the principles of its measurement with US shear wave.
Results
Lines 132-133 «The Spearman’s rank correlation coefficient showed that dependencies between masseter muscle stiffness and temporalis muscle stiffness were statistically significant.» The presence of correlation does not automatically imply a “dependency” between variables. You should rephrase the sentences because it is unclear. Actually, I don’t’ understand the necessity to determine a correlation between the masseter and temporalis. Maybe it should be better justified in the text, to be understandable to the readers.
I find that for a more accurate description of the results, not only the absolute values of the stiffness but also the variation relative to the baseline should be presented.
Discussion
Lines 158-171 This paragraph presents an explanation of how food consistency can influence the mastication patterns and this, in turn, influences muscle work. However, in the experiments, the food consistency was the same for all the participants, then the food consistency cannot be considered as a factor for the different patterns. this whole paragraph seems disconnected from the study aim.
There is no discussion of the physiological causes for the stiffness increase and the possible explanation of the partial drop in stiffness.
Author Response
Reviewer 1
COMMENT 1: Overall comments - The authors present an experimental study that assessed by means of shear wave US the stiffness of masticatory muscles before and after chewing gum. The paper gives interesting information about the clinical application of this type of measurement in the field of TMD. It might be a valuable reference for other studies. The paper is clearly structured and easy to follow. However, I think that some content issues should be addressed in order to improve its quality. Namely, I think that the description of the US method principles must be improved, as well as the discussion should be better structured and be more adherent to the presented results and their relevance to the field and their limitations.
RESPONSE 1: We are pleased to send you the revised version of the article entitled “Monitoring of changes in masticatory muscle stiffness changes after gum chewing using shear-wave elastography” for consideration by the Journal of Clinical Medicine. We would like to thank you for your interest in this work and for all the comments and advice you have given us. You will find below a list of the changes made to the manuscript showing that all your remarks have been considered.
COMMENT 2: Introduction - Lines-69-71 Citations should be added to corroborate these statements.
RESPONSE 2: Thank you for this remark. We have added the following references: Woźniak et al. 2015; Sarwono et al. 2019 and Lee et al. 2021.
COMMENT 3: Material and methods - General remark: the order of the experimental session is confused. The recruitment criteria at first, and only then explain the experimental procedure. Consider restructuring.
RESPONSE 3: We have improved the methods section. The structure is now as follow: participant, study procedure, description of the device, and ethical statement.
COMMENT 4: Lines 75-77 “Examinations were carried out in the same conditions, with the same chewing gum and settings to diminish the impact of other factors on the outcomes.» Maybe the factors that could influence the outcomes should be named.
RESPONSE 4: Thank you for this comment. We meant to diminish the impact of the above-mentioned factors. So, we have replaced the phrase “other factors” with “these factors”.
COMMENT 5: Lines 78-79 There is no clear explanation of the chewing protocol. Any indication on the chewing side? Was kept free? the subject’s favorite chewing side was considered? This can be also considered a confounding factor for the results. It would also be interesting to relate the chewing pattern to the chewing side and the changes in the stiffness of the muscles on that side.
RESPONSE 5: This topic is interesting. We thank you for asking these questions. The participants could chew the gum according to their preference. We have not taken into account the subject’s favourite chewing side. Focusing on the chewing pattern would require to supplement methodology with additional methods. As we did not consider the chewing side in this study, we mentioned this factor in the limitations of the study.
Also, to address this comment further, we have compared obtained results between left and right measurements. All differences except one are not significant. Such a result would suggest that the chewing side might be not essential. However, this study was not designed to explore this task, and further studies are needed.
COMMENT 6: Lines 102-103 how was the position of the region of interest defined and how repeatability ensured? Since the measurement is repeated before and after the chewing, the repositioning of the probe and the selection of the ROI are crucial.
RESPONSE 6: This is a crucial issue of the examination of the masseter muscle. Thank you for this comment. We have been gaining experience for a long period of time, which resulted in already published papers (e.g., Olchowy et al. Dentomaxillofac Radiol, 2020; Olchowy et al. Pain Res Manag, 2020), while other papers are currently under review.
We have noticed that selecting an area for measurements in the middle of the masseter belly is easy and repeatable. Also, part of the temporalis muscle just above the zygomatic arch is easy to identify. However, to ensure the accuracy of repeated measurements, we used a linear skin mark pointing to an exact probe location and orientation. We have added additional information to the methods.
COMMENT 7: Furthermore, the whole analysis of the elasticity is missing. It would be helpful to include how the elasticity is calculated by the instruments and at least the principles of its measurement with US shear wave.
RESPONSE 7: Shear wave elastography is already proven and a widely accepted method of measuring hardness/elasticity of soft tissues. The thyroid, breast and liver pathologies have hundreds, if not thousands of publications, discussing technical details over the last decade. On this stage of adoption of shear wave elastography in clinical practice, it would be maybe unnecessary and beyond the scope of our study to repeat this information. Furthermore, results are calculated by software without any action required from the operator. The published literature shows that the measurements using this equipment are reliable and repeatable. Kilopascals represent the hardness of a tissue. Measurements are very accurate - up to one-tenth of kilopascal. Shear wave elastography is very sensitive for even slight changes in hardness/elasticity. Using the term “hardness” maybe better represents what we measure; however, scientific literature uses the term elasticity more often.
COMMENT 8: Results - Lines 132-133 «The Spearman’s rank correlation coefficient showed that dependencies between masseter muscle stiffness and temporalis muscle stiffness were statistically significant.» The presence of correlation does not automatically imply a “dependency” between variables. You should rephrase the sentences because it is unclear. Actually, I don’t’ understand the necessity to determine a correlation between the masseter and temporalis. Maybe it should be better justified in the text, to be understandable to the readers.
RESPONSE 8: We agree. A correlation describes a point in time, but not dependency. We have replaced the word ‘dependency’ with ‘relationship’.
COMMENT 9: I find that for a more accurate description of the results, not only the absolute values of the stiffness but also the variation relative to the baseline should be presented.
RESPONSE 9: We have calculated the additional values and added a section in the results to read as follows:
For the masseter muscle, the elasticity increased by 1.05±0.53 kPa at the first measurement from the baseline and then dropped by 0.89±0.58 kPa at the second measurement. The elasticity of the temporalis muscle increased by 0.64±0.77 kPa after chewing and dropped by 0.19±0.42 kPa after relaxation.
COMMENT 10: Discussion - Lines 158-171 This paragraph presents an explanation of how food consistency can influence the mastication patterns and this, in turn, influences muscle work. However, in the experiments, the food consistency was the same for all the participants, then the food consistency cannot be considered as a factor for the different patterns. this whole paragraph seems disconnected from the study aim.
RESPONSE 10: Thank you for bringing this to our attention. We have deleted this paragraph. In turn, more interesting information was added to the discussion.
COMMENT 11: There is no discussion of the physiological causes for the stiffness increase and the possible explanation of the partial drop in stiffness.
RESPONSE 11: Thank you for pointing this out. We have added 2 paragraphs to the discussion and deleted a paragraph on chewing patterns. Now, the discussion gained more value and a better flow.
Reviewer 2 Report
The authors demonstrate that the intensive exercise affects the stiffness of the masticatory muscles measured with shear-wave elastography.
Please add about the pointed out matter.
1. Intra-rater reliability must be shown in this manuscript.
2. Please describe the measurement probe position and the measurement echo image in the figure.
Author Response
Reviewer 2
COMMENT 1: The authors demonstrate that the intensive exercise affects the stiffness of the masticatory muscles measured with shear-wave elastography.
RESPONSE 1: We are pleased to send you the revised version of the article entitled “Monitoring of changes in masticatory muscle stiffness changes after gum chewing using shear-wave elastography” for consideration by the Journal of Clinical Medicine. Thank you very much for your time and the clear and valuable suggestions you have raised. Please find below answers to your comments which are also included in the revised manuscript.
COMMENT 2: Please add about the pointed out matter.
- Intra-rater reliability must be shown in this manuscript.
RESPONSE 2: We would like to thank you for this comment is it relates to a very important issue of the repeatability of results. Intra-rater reliability is one of the measures of agreement of examinations performed by a single operator. This study, however, has a different goal. In this study, we did not perform enough measurements in a similar environment to be able to calculate intra-rater reliability. Measurements in this study were carried out by an experienced radiologist, which increases the accuracy of results.
In another publication, we showed that the repeatability of shear wave elastography is very high. This paper was accepted for publication, but it is not available yet. We can add a citation when it appears online and receives DOI. Furthermore, this method was validated for other tissues with results available in the literature. For this reason, we assumed that the method is reliable and accurate with repeatable results.
COMMENT 3: 2. Please describe the measurement probe position and the measurement echo image in the figure.
RESPONSE 3: To address this comment, we have improved the description of the methods. It is now as follows:
The probe was placed parallel to the longitudinal axis of the masseter muscle in the widest part (the midpoint level), of the masseter muscle belly. Regarding the temporalis muscle, the probe was placed in the temporal area longitudinally to the fibres of the temporalis muscle just above the zygomatic arch. Based on our previous experience, such locations on both muscles can be easily identified and provide repeatable results. However, to ensure the accuracy of repeated measurements, we used a linear skin mark pointing to an exact probe location and orientation.
Round 2
Reviewer 1 Report
The authors have addressed the suggested modifications and improved the quality of the manuscript.